# Rethinking the Role of Hyperparameter Tuning in Optimizer Benchmarking

**Yuanhao Xiong[1], Xuanqing Liu[1], Li-Cheng Lan[1], Yang You[2], Si Si[3], Cho-Jui Hsieh[1]**
[1]Department of Computer Science, UCLA
[2]Department of Computer Science, National University of Singapore
[3]Google Research
{yhxiong,xqliu,lclan}@cs.ucla.edu, youy@comp.nus.edu.sg
sisidaisy@google.com, chohsieh@cs.ucla.edu

## Abstract

Many optimizers have been proposed for training deep neural networks, and they often have multiple hyperparameters, which make it tricky to benchmark their performance. In this work, we propose a new benchmarking protocol to evaluate both end-to-end efficiency (training a model from scratch without knowing the best hyperparameter configuration) and data-addition training efficiency (the previously selected hyperparameters are used for periodically re-training the model with newly collected data). For end-to-end efficiency, unlike previous work that assumes random hyperparameter tuning, which may over-emphasize the tuning time, we propose to evaluate with a bandit hyperparameter tuning strategy. For data-addition training, we design a new protocol for assessing the hyperparameter sensitivity to data shift. We then apply the proposed benchmarking framework to 7 optimizers on various tasks, including computer vision, natural language processing, reinforcement learning, and graph mining. Our results show that there is no clear winner across all the tasks.

## 1 Introduction

Due to the enormous data size and non-convexity, stochastic optimization algorithms have become widely used in training deep neural networks. In addition to Stochastic Gradient Descent (SGD) [27], many variations such as Adagrad [11], RMSprop [34] and Adam [17] have been proposed with better performance. Unlike classical and hyperparameter free optimizers such as gradient descent and Newton's method[1], stochastic optimizers often hold multiple hyperparameters including learning rate and momentum coefficients. Those hyperparameters are critical not only to the training speed, but also to the final performance, and are often hard to tune.

It is thus non-trivial to benchmark and compare optimizers in deep neural network training. And a benchmarking mechanism that focuses on the performance under best hyperparameters could lead to a false sense of improvement when developing new optimizers without considering tuning efforts. In this paper, we aim to rethink the role of hyperparameter tuning in benchmarking optimizers and develop new benchmarking protocols to reflect their performance in practical tasks better. We then benchmark seven recently proposed and widely used optimizers and study their performance on a wide range of tasks with our proposed protocols. In the following, we will first briefly review the two existing benchmarking protocols, discuss their pros and cons, and then introduce our contributions.

---

[1]The step sizes of gradient descent and Newton's method can be automatically adjusted by a line search procedure [24].

Submitted to the 35th Conference on Neural Information Processing Systems (NeurIPS 2021) Track on Datasets and Benchmarks. Do not distribute.

**Benchmarking performance under the best hyperparameters.** A majority of previous benchmarks and comparisons on optimizers are based on the best hyperparameters. Wilson et al. [36] and Shah et al. [31] made a comparison of SGD-based methods against adaptive ones under their best hyperparameter configurations. They found that SGD can outperform adaptive methods on several datasets under careful tuning. Most of the benchmarking frameworks for ML training also assume knowing the best hyperparameters for optimizers [29, 9, 42]. Also, the popular MLPerf benchmark evaluated the performance of optimizers under the best hyperparameter. It showed that ImageNet and BERT could be trained in 1 minute using the combination of good optimizers, good hyperparameters, and thousands of accelerators.

Despite each optimizer's peak performance being evaluated, benchmarking under the best hyperparameters makes the comparison between optimizers unreliable and fails to reflect their practical performance. First, the assumption of knowing the best hyperparameter is unrealistic. In practice, it requires a lot of tuning efforts to find the best hyperparameter, and the tuning efficiency varies greatly for different optimizers. It is also tricky to define the "best hyperparameter", which depends on the hyperparameter searching range and grids. Further, since many of these optimizers are sensitive to hyperparameters, some improvements reported for new optimizers may come from insufficient tuning for previous work.

**Benchmarking performance with random hyperparameter search.** It has been pointed out in several papers that tuning hyperparameter needs to be considered in evaluating optimizers [29, 2], but having a formal evaluation protocol on this topic is non-trivial. Only recently, two papers Choi et al. [8] and Sivaprasad et al. [33] take hyperparameter tuning time into account when comparing SGD with Adam/Adagrad. However, their comparisons among optimizers are conducted on random hyperparameter search. We argue that these comparisons could over-emphasize the role of hyperparameter tuning, which could lead to a pessimistic and impractical performance benchmarking for optimizers. This is due to the following reasons: First, in the random search comparison, each bad hyperparameter has to run fully (e.g., 200 epochs). In practice, a user can always stop the program early for bad hyperparameters if having a limited time budget. For instance, if the learning rate for SGD is too large, a user can easily observe that SGD diverges in a few iterations and directly stops the current job. Therefore, the random search hypothesis will over-emphasize the role of hyperparameter tuning and does not align with a real user's practical efficiency. Second, the performance of the best hyperparameter is crucial for many applications. For example, in many real-world applications, we need to re-train the model every day or every week with newly added data. So the best hyperparameter selected in the beginning might benefit all these re-train tasks rather than searching parameters from scratch. In addition, due to the expensive random search, random search based evaluation often focuses on the low-accuracy region[2], while practically we care about the performance for getting reasonably good accuracy.

**Our contributions.** Given that hyperparameter tuning is either under-emphasized (assuming the best hyperparameters) or over-emphasize (assuming random search) in existing benchmarking protocols and comparisons, we develop **new evaluation protocols** to compare optimizers to reflect the real use cases better. Our evaluation framework includes two protocols. First, to evaluate the **end-to-end training efficiency** for a user to train the best model from scratch, we develop an efficient evaluation protocol to compare the accuracy obtained under various time budgets, including the hyperparameter tuning time. Instead of using the random search algorithm, we adopt the Hyperband [19] algorithm for hyperparameter tuning since it can stop early for bad configurations and better reflect the real running time required by a user. Further, we also propose to evaluate the **data addition training efficiency** for a user to re-train the model with some newly added training data, with the knowledge of the best hyperparameter tuned in the previous training set.

Based on the proposed evaluation protocols, we **study how much progress has recently proposed algorithms made compared with SGD or Adam**. Note that most of the recent proposed optimizers have shown outperforming SGD and Adam under the best hyperparameters for some particular tasks, but it is not clear whether the improvements are still significant when considering hyper-parameter tuning, and across various tasks. To this end, we conduct comprehensive experiments comparing state-of-the-art training algorithms, including SGD [27], Adam [17], RAdam [20], Yogi [40], LARS [37], LAMB [38], and Lookahead [41], on a variety of training tasks including image classification, generated adversarial networks (GANs), sentence classification (BERT fine-tuning), reinforcement learning and graph neural network training. Our main conclusions are: 1) On CIFAR-10 and CIFAR-

---

[2]For instance, Sivaprasad et al. [33] only reaches $< 50\%$ accuracy in their CIFAR-100 comparisons.

100, all the optimizers including SGD are competitive. 2) Adaptive methods are generally better on more complex tasks (NLP, GCN, RL). 3) There is no clear winner among adaptive methods. Although RAdam is more stable than Adam across tasks, Adam is still a very competitive baseline even compared with recently proposed methods.

# 2 Related Work

**Optimizers.** Properties of deep learning make it natural to apply stochastic first order methods, such as Stochastic Gradient Descent (SGD) [27]. Several issues such as a zig-zag training trajectory and a uniform learning rate have been exposed, and researchers have then drawn extensive attention to modify the existing SGD for improvement. Along this line of work, tremendous progresses have been made including SGDM [25], Adagrad [11], RMSProp [34], and Adam [17]. These methods utilize momentums to stabilize and speed up training procedures. In particular, Adam is regarded as the default algorithm due to its outstanding compatibility. Then variants such as Amsgrad [26], Adabound [21], Yogi [40], and RAdam [20] have been proposed to resolve different drawbacks of Adam. Meanwhile, the requirement of large batch training has inspired the development of LARS [37] and LAMB [38]. Moreover, Zhang et al. [41] has put forward Lookahead to boost optimization performance by iteratively updating two sets of weights. Layer-wise adaptive moments (NovoGrad [14]) and sharpness-aware minimization (SAM [13] and SALR [39]) have also been proposed to improve optimization in deep learning. With the rapid developed of optimization algorithms in deep learning, it is important to benchmark them with a fair protocol. DeepOBS [29] is one of a deep learning optimizer benchmark suite and Schmidt et al. [28] further conduct a larger scaled evaluation with 1920 configurations of different hyperparameter settings. However, these papers only focus on the final performance and neglect the importance of hyperparameter tuning effort. Although Sivaprasad et al. [33] take hyperparameter tuning into account, random search as the HPO method might not be the proper choice to reflect the impact of hyperparameter tuning fairly, which is discussed in detail in Section 3.1.

**Hyperparameter tuning methods.** Random search and grid search [4] are two basic hyperparameter tuning methods in the literature. However, the inefficiency of these methods stimulates the development of more advanced search strategies. Bayesian optimization methods including Bergstra et al. [5] and Hutter et al. [16] accelerate random search by fitting a black-box function of hyperparameter and the expected objective to adaptively guide the search direction. Results in a recent competition [35] have pointed out that Bayesian optimization is superior to random search in hyperparameter tuning. Parallel to this line of work, Hyperband [19] focuses on reducing evaluation cost for each configuration and early terminates relatively worse trials. Falkner et al. [12] proposes BOHB to combine the benefits of both Bayesian Optimization and Hyperband. All these methods still require huge computation resources. A recent work [22] has tried to obtain a list of potential hyperparameters by meta-learning from thousands of representative tasks. We strike a balance between effectiveness and computing cost and leverage Hyperband in our evaluation protocol to compare a wider range of optimizers.

# 3 Proposed Evaluation Protocols

In this section, we introduce the proposed evaluation framework for optimizers. We consider two evaluation protocols, each corresponding to an important training scenario:

- **Scenario I (End-to-end training)**: This is the general training scenario, where a user is given an unfamiliar optimizer and task, the goal is to achieve the best validation performance after several trials and errors. In this case, the evaluation needs to include hyperparameter tuning time. We develop an efficiency evaluation protocol to compare various optimizers in terms of CPE and peak performance.
- **Scenario II (Data-addition training)**: This is another useful scenario encountered in many applications, where the same model needs to be retrained regularly after collecting some fresh data. In this case, a naive solution is to reuse the previously optimal hyperparameters and retrain the model. However, since the distribution is shifted, the result depends on the sensitivity to that shift.

We describe the detailed evaluation protocol for each setting in the following subsections.

## 3.1 End-to-end Training Evaluation Protocol

Before introducing our evaluation protocol for Scenario I, we first formally define the concept of optimizer and its hyperparameters.

**Definition 1.** *An optimizer is employed to solve a minimization problem $\min_\theta \mathcal{L}(\theta)$ and can be defined by a tuple $o \in \mathcal{O} = (\mathcal{U}, \Omega)$, where $\mathcal{O}$ contains all types of optimizers. $\mathcal{U}$ is a specific update rule and $\Omega = (\omega_1, \ldots, \omega_N) \in \mathbb{R}^N$ represents a vector of $N$ hyperparameters. Search space of these hyperparameters is denoted by $\mathcal{F}$. Given an initial parameter value $\theta_0$, together with a trajectory of optimization procedure $H_t = \{\theta_s, \mathcal{L}(\theta_s), \nabla \mathcal{L}(\theta_s)\}$, the optimizer updates $\theta$ by*

$$\theta_{t+1} = \mathcal{U}(H_t, \Omega).$$

We aim to evaluate the end-to-end time for a user to get the best model, including the hyperparameter tuning time. A recent work [33] assumes that a user conducts random search for finding the best hyperparameter setting. Still, we argue that the random search procedure will *over-emphasize* the importance of hyperparameters when tuning is considered — it assumes a user never stops the training even if they observe divergence or bad results in the initial training phase, which is unrealistic.

Figure 1 illustrates why random search might not lead to a fair comparison of optimizers. In Figure 1, we are given two optimizers, A and B, and their corresponding loss w.r.t. hyperparameter. According to Sivaprasad et al. [33], optimizer B is considered better than optimizer A under a constrained budget since most regions of the hyperparameter space of B outperforms A. For instance, suppose we randomly sample the same hyperparamter setting for A and B. The final config $\omega_r^*(B)$ found under this strategy can have a lower expected loss than that of $\omega_r^*(A)$, as shown in Figure 1a. However, there exists a more practical search strategy which can invalidate this statement with the assumption of a limited searching budget: a user can early terminate a configuration trial when trapped in bad results or diverging. Hence, we can observe in Figure 1b that for optimizer A, this strategy early-stops many configurations and only allow a limited number of trials to explore to the deeper stage. Therefore, the bad hyperparameters will not affect the overall efficiency of optimizer A too much. In contrast, for optimizer B, performances of different hyperparameters are relatively satisfactory and hard to distinguish, resulting in similar and long termination time for each trial. Therefore, it may be easier for a practical search strategy $p$ to find the best configuration $\omega_p^*(A)$ of optimizer A than $\omega_p^*(B)$, given the same constrained budget.

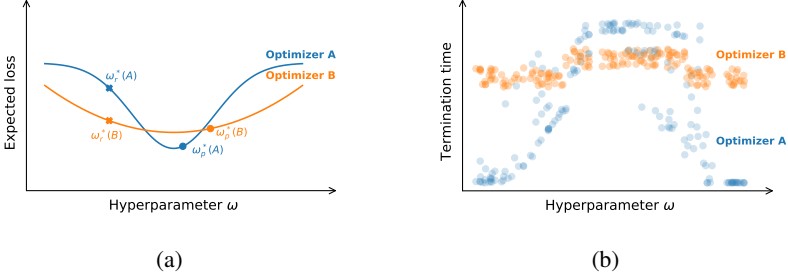

|     |     |
|:---:|:---:|
| (a) | (b) |

Figure 1: An illustration example showing that different hyperparamter tuning methods are likely to affect comparison of optimizers. Optimizer A is more sensitive to hyperparamters than optimizers B, but it may be prefered if bad hyperparameters can be terminated in the early stage.

This example suggests that random search may over-emphasize the parameter sensitivity when benchmarking optimizers. To better reflect a practical hyperparameter tuning scenario, our evaluation assumes a user applies **Hyperband** [19], a simple but effective hyperparameter tuning scheme to get the best model. Hyperband formulates hyperparameter optimization as a unique bandit problem. It accelerates random search through adaptive resource allocation and early-stopping, as demonstrated in Figure 1b. Compared with its more complicated counterparts such as BOHB [12], Hyperband requires less computing resources and performs similarly within a constrained budget. The algorithm is presented in Appendix A.

To validate the effectiveness of Hyperband, we make a comparison among different HPO algorithms. In detail, we conduct hyperparameter tuning for image classification on CIFAR10, given 10 learning rate configurations of SGD in the grid $[1.0 \times 10^{-8}, 1.0 \times 10^{-7}, 1.0 \times 10^{-6}, \ldots, 10]$. The budget for each configuration is 200 epochs. We consider following HPO methods: Hyperband, random search,

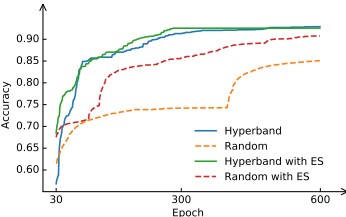

Figure 2: Hyperband tuning used in our evaluation protocol outperforms random search consistently.

random search with an early stopping (ES) strategy in Sivaprasad et al. [33], and Hyperband with ES.
In Figure 2, we plot corresponding performance for these methods. We find that Hyperband outperforms random search consistently, while random search tends to trap in suboptimal configurations even though early stopping can mitigate this issue to some extent. This finding shows the advantage of Hyperband over random search regardless of early stopping, and justifies the use of Hyperband in optimizer benchmarking.

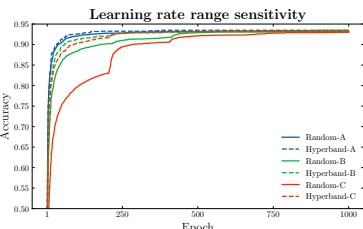

Figure 3: Sensitivity to search space range.

Besides, the adaptive terminating strategy makes Hyperband less sensitive to the range of search space than other hyperparameter tuning methods like random search. We use Hyperband and random search to tune learning rate of SGD for a classification problem on CIFAR10 with three different search spaces: $[10^{-3}, 10^{-1}]$(A), $[10^{-4}, 10^{0}]$(B), and $[10^{-5}, 10^{1}]$(C). We sample $50$ configurations based on log-uniform distribution within each range. As shown in Figure 3, random search suffers great performance degradation when the range becomes larger from A to C, while Hyperband performs consistently well on three ranges.

With Hyperband incorporated in end-to-end training, we assume that each configuration is run sequentially and record the best performance obtained at time step $t$ as $P_t$. Specifically, $P_t$ represents the evaluation metric for each task, e.g., accuracy for image classification and return for reinforcement learning. $\{P_t\}_{t=1}^{T}$ forms a trajectory for plotting learning curves on test set like Figure 5. Although it is intuitive to observe the performance of different optimizers according to such figures, summarizing a learning curve into a quantifiable, scalar value can be more insightful for evaluation. Thus, as shown in Eq. 1, we use $\lambda$-tunability defined in [33] to further measure the performance of optimizers:

$$\lambda\text{-tunability} = \sum_{t=1}^{T} \lambda_t \cdot P_t (\sum_t \lambda_t = 1 \text{ and } \forall_t \lambda_t > 0). \tag{1}$$

One intuitive way is to set $\lambda_t = \mathbf{1}_{t=T}$ with $\lambda_T = 1$ and $\lambda_t = 0$ for the rest to determine which optimizer can reach the best performance after the whole training procedure. However, merely considering the peak performance is not a good guidance on the choice of optimizers. In practice, we tend to take into account the complete trajectory and exert more emphasis on the early stage. Thus, we employ the Cumulative Performance-Early weighting scheme where $\lambda_t \propto (T - i)$, to compute $\lambda$-tunablity instead of the extreme assignment $\lambda_t = \mathbf{1}_{t=T}$. The value obtained is termed as *CPE*.

We present our evaluation protocol in Algorithm 1. As we can see, end-to-end training with hyperparameter optimization is conducted for various optimizers on the given task. The trajectory $\{P_t\}_{t=1}^{T}$ is recorded to compute the peak performance as well as *CPE* value. Note that the procedure is repeated $M$ times to obtain a reliable result. We use $M = 3$ in all experiments.

**Algorithm 1** End-to-End Efficiency Evaluation Protocol

---

**Input:** A set of optimizers $\mathcal{O} = \{o : o = (\mathcal{U}, \Omega)\}$, task $a \in \mathcal{A}$, feasible search space $\mathcal{F}$

1: **for** $o \in \mathcal{O}$ **do**
2:     **for** $i = 1$ **to** $M$ **do**
3:         Conduct hyperparameter search in $\mathcal{F}$ with the optimizer $o$ using HyperBand on $a$
4:         Record the performance trajectory $\{P_t\}_{t=1}^{T}$ explored by HyperBand
5:         Calculate the peak performance and *CPE* by Eq. 1
6:     **end for**
7:     Average peak and *CPE* values over $M$ repetitions for the optimizer $o$
8: **end for**
9: Evaluate optimizers based on their peak and *CPE* values

---

Moreover, we can further accelerate our evaluation protocol. The basic idea is to keep a library of trajectories for different hyperparameter settings. We first sample a list of configurations to be evaluated. In each repetition, we sample required configurations from the list to conduct one Hyperband running. During the simulation of Hyperband, we just retrieve the value if the desired epoch of current configuration is contained in the library. Otherwise, we run this configuration based on Hyperband, and store the piece of the trajectory to the library. More details of the algorithm can be found in Appendix D.

### 3.2 Data-addition Training Evaluation Protocol

In Scenario II, we assume that there's a service (e.g., a search or recommendation engine) which is being re-trained periodically with some newly added training data. One may argue that an online learning algorithm should be used in this case, but in practice online learning is unstable and industries still prefer this periodically retraining scheme which is more stable.

In this scenario, once the best hyperparameters were chosen in the beginning, we can reuse them for every training, so no hyperparameter tuning is required and the performance (including both efficiency and test accuracy) under the best hyperparameter becomes important. However, an implicit assumption made in this process is that *"the best hyperparameter will still work when the training task slightly changes"*. This can be viewed as transferability of hyperparameters for a particular optimizer, and our second evaluation protocol aims to evaluate this practical scenario.

We simulate data-addition training with all classification tasks, and the evaluation protocol works as follows: 1) Extract a subset $\mathcal{D}_\delta$ containing partial training data from the original full dataset $\mathcal{D}$ with a small ratio $\delta$; 2) Conduct a hyperparameter search on $\mathcal{D}_\delta$ to find the best setting under this scenario; 3) Use these hyperparameters to train the model on the complete dataset; 4) Observe the potential change of the ranking of various optimizers before and after data addition. For step 4) when comparing different optimizers, we will plot the training curve in the full-data training stage in Section 4, and also summarize the training curve using the *CPE* value. The detailed evaluation protocol is described in Algorithm 2.

---

**Algorithm 2** Data-Addition Training Evaluation Protocol

---

**Input:** A set of optimizers $\mathcal{O} = \{o : o = (\mathcal{U}, \Omega)\}$, task $a \in \mathcal{A}$ with a full dataset $\mathcal{D}$, a split ratio $\delta$

1: **for** $o \in \mathcal{O}$ **do**
2:     **for** $i = 1$ **to** $M$ **do**
3:         Conduct hyperparameter search with the optimizer $o$ using Hyperband on $a$ with a partial dataset $\mathcal{D}_\delta$, and record the best hyperparameter setting $\Omega_{\text{partial}}$ found under this scenario
4:         Apply the optimizer with $\Omega_{\text{partial}}$ on $\mathcal{D}_\delta$ and $\mathcal{D}$, then save the training curves
5:     **end for**
6:     Average training curves of $o$ over $M$ repetitions to compute *CPE*
7: **end for**
8: Compare performance of different optimizers under data-addition training

---

## 4 Experimental Results

**Optimizers to be evaluated.** As shown in Table 1, we consider 7 optimizers including non-adaptive methods using only the first-order momentum, and adaptive methods considering both first-order and second-order momentum. We also provide lists of tunable hyperparameters for different optimizers in Table 1. Moreover, we consider following two combinations of tunable hyperparameters to better investigate the performance of different optimizers: **a)** only tuning initial learning rate with the others set to default values and **b)** tuning a full list of hyperparameters. A detailed description of optimizers as well as default values and search range of these hyperparameters can be found in Appendix D. We adopt a unified search space for a fair comparison following Metz et al. [22], to eliminate biases of specific ranges for different optimizers. The tuning budget of Hyperband is determined by three items: maximum resource (in this paper we use epoch) per configuration $R$, reduction factor $\eta$, and number of configurations $n_c$. According to Li et al. [19], a single Hyperband execution contains $n_s = \lfloor \log_\eta(R) \rfloor + 1$ of SuccessiveHalving, each referred to as a bracket. These brackets take strategies from least to most aggressive early-stopping, and each one is designed to use approximately $B = R \cdot n_s$ resources, leading to a finite total budget. The number of randomly sampled configurations in one Hyperband run is also fixed and grows with $R$. Then given $R$ and $\eta$, $n_c$ determines the repetition times of Hyperband. We set $\eta = 3$ as this default value performs consistently well, and $R$ to a value which each task usually takes for a complete run. For $n_c$, it is assigned as what is required for a single Hyperband execution for all tasks, except for BERT fine-tuning, where a larger number of configurations is necessary due to a relatively small $R$. In Appendix D, we give assigned values of $R$, $\eta$, and $n_c$ for each task.

Table 1: Optimizers to be evaluated with their tunable hyperparameters. Specifically, $\alpha_0$ represents the initial learning rate. $\mu$ is the decay factor of the first-order momentum for non-adaptive methods while $\beta_1$ and $\beta_2$ are coefficients to compute the running averages of first-order and second-order momentums. $\epsilon$ is a small scalar used to prevent division by 0.

| Optimizer | | Hyperparameter |
|---|---|---|
| Non-adaptive | SGD | $\alpha_0, \mu$ |
| | LARS | $\alpha_0, \mu, \epsilon$ |
| Adaptive | Adam, RAdam, Yogi Lookahead, LAMB | $\alpha_0, \beta_1, \beta_2, \epsilon$ |

Table 2: Tasks for benchmarking optimizers. Details are provided in Appendix C.

| Domain | Task | Metric | Model | Dataset |
|---|---|---|---|---|
| Computer Vision | Image Classification | Accuracy | ResNet-50 | CIFAR10 CIFAR100 |
| | VAE | NLL | CNN Autoencoder | CelebA |
| | GAN | FID | SNGAN network | CIFAR10 |
| NLP | GLUE benchmark | Accuracy | RoBERTa-base | MRPC |
| Graph network training | Node labeling | F1 score | Cluster-GCN | PPI |
| Reinforcement Learning | Walker2d-v3 | Return | PPO | $\times$ |

Table 3: CPE for different optimizers on benchmarking tasks. The best performance is highlighted in bold and blue and results within the 1% range of the best are emphasized in bold only.

| Optimizer | CIFAR10 (%) ↑ (classification) | CIFAR100 (%) ↑ (classification) | CelebA ↓ (VAE) | MRPC (%) ↑ (NLP) | PPI ↑ (GCN) | Walker2d-v3 ↑ (RL) |
|---|---|---|---|---|---|---|
| *Tune learning rate only:* | | | | | | |
| SGD | 88.87 ± 0.23 | **66.85 ± 0.12** | 0.1430 ± 0.0038 | 69.90 ± 0.69 | 76.77 ± 0.08 | 2795 ± 275 |
| Adam | **90.42 ± 0.10** | 65.88 ± 0.23 | **0.1356 ± 0.0001** | **84.90 ± 0.72** | **95.08 ± 0.01** | 3822 ± 78 |
| RAdam | **90.29 ± 0.11** | 66.41 ± 0.15 | **0.1362 ± 0.0001** | **85.41 ± 1.45** | 94.10 ± 0.04 | 3879 ± 201 |
| Yogi | **90.42 ± 0.04** | **67.37 ± 0.50** | 0.1371 ± 0.0004 | 70.19 ± 0.90 | 93.39 ± 0.02 | **4132 ± 205** |
| LARS | **90.25 ± 0.07** | **67.48 ± 0.04** | 0.1367 ± 0.0002 | 69.97 ± 0.54 | 93.79 ± 0.01 | 2986 ± 105 |
| LAMB | **90.19 ± 0.08** | 65.08 ± 0.06 | 0.1358 ± 0.0003 | 82.23 ± 1.49 | 87.79 ± 0.07 | 3401 ± 235 |
| Lookahead | **90.60 ± 0.06** | 65.60 ± 0.07 | 0.1358 ± 0.0004 | 72.99 ± 1.33 | **94.69 ± 0.02** | **4141 ± 264** |
| *Tune every hyperparameter:* | | | | | | |
| SGD | **90.20 ± 0.16** | **67.36 ± 0.10** | 0.1407 ± 0.0011 | 71.53 ± 1.21 | 94.64 ± 0.01 | 2978 ± 91 |
| Adam | 89.27 ± 1.40 | **67.57 ± 0.23** | 0.1389 ± 0.0002 | **85.23 ± 1.44** | 92.62 ± 0.04 | **4080 ± 459** |
| RAdam | **90.14 ± 0.44** | 66.90 ± 0.05 | **0.1366 ± 0.0006** | 84.32 ± 1.91 | 93.05 ± 0.04 | 3813 ± 103 |
| Yogi | 89.83 ± 0.21 | **67.65 ± 0.08** | 0.1401 ± 0.0019 | 68.42 ± 1.02 | 88.94 ± 0.11 | 3778 ± 249 |
| LARS | **90.42 ± 0.20** | **67.78 ± 0.28** | 0.1375 ± 0.0005 | 77.40 ± 3.09 | **96.34 ± 0.01** | 2728 ± 136 |
| LAMB | **90.27 ± 0.40** | 65.59 ± 0.03 | 0.1382 ± 0.0001 | **84.66 ± 2.61** | 93.18 ± 0.05 | 2935 ± 57 |
| Lookahead | **90.44 ± 0.11** | 66.46 ± 0.45 | **0.1360 ± 0.0005** | 79.05 ± 2.99 | 94.30 ± 0.04 | 3786 ± 137 |

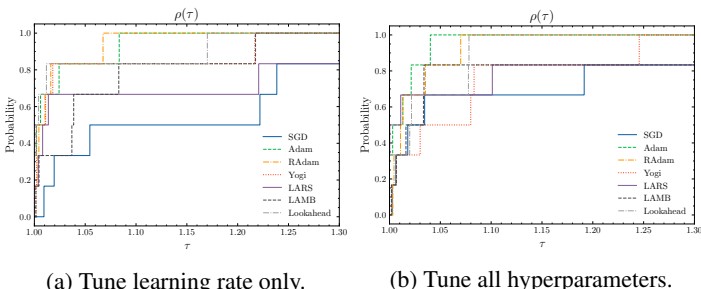

(a) Tune learning rate only.  (b) Tune all hyperparameters.

Figure 4: Performance profile in the range $[1.0, 1.3]$.

**Tasks for benchmarking.** For a comprehensive and reliable assessment of optimizers, we consider a wide range of tasks in different domains. When evaluating end-to-end training efficiency, we implement our protocol on tasks covering several popular and promising applications in Table 2. Apart from common tasks in computer vision and natural language processing, we introduce two extra tasks in graph neural network training and reinforcement learning. For simplicity, we will use the dataset to represent each task in our subsequent tables of experimental results. (For the reinforcement learning task, we just use the environment name.) The detailed settings and parameters for each task can be found in Appendix C.

## 4.1 End-to-end efficiency (Secnario I)

Table 4: CPE of different optimizers computed under curves trained with $\Omega_{\text{partial}}$ on four full datasets.

| Optimizer | CIFAR10 (%)↑ | CIFAR100 (%)↑ | MRPC (%)↑ | PPI↑ |
|---|---|---|---|---|
| SGD | **90.04 ± 0.16** | **67.91 ± 0.23** | 66.62 ± 3.47 | 66.830 ± 0.010 |
| Adam | **90.52 ± 0.03** | 67.04 ± 0.27 | 73.13 ± 1.16 | **70.420 ± 0.007** |
| RAdam | **90.30 ± 0.14** | 67.06 ± 0.17 | **79.01 ± 3.10** | **70.840 ± 0.010** |
| Yogi | **89.63 ± 0.39** | **67.58 ± 0.19** | 68.40 ± 1.68 | 67.990 ± 0.003 |
| LARS | **90.17 ± 0.13** | **67.29 ± 0.14** | 64.43 ± 2.72 | 68.400 ± 0.005 |
| LAMB | **90.51 ± 0.07** | 66.13 ± 0.02 | **78.94 ± 1.25** | 70.110 ± 0.008 |
| Lookahead | 88.36 ± 0.06 | 67.10 ± 0.31 | 68.81 ± 1.22 | 69.710 ± 0.003 |

To evaluate end-to-end training efficiency, we adopt the protocol in Algorithm 1. Specifically, we record the average training trajectory with Hyperband $\{P_t\}_{t=1}^{T}$ for each optimizer on benchmarking tasks, where $P_t$ is the evaluation metric for each task (e.g., accuracy, return). We visualize these trajectories in Figure 5a and 5b for CIFAR100, and calculate *CPE* in Table 3. Complete results of trajectories and peak performances for all tasks can be found in Appendix E. Besides, in Eq. 2 we compute *performance ratio* $r_{o,a}$ for each optimizer and each task, and then utilize the distribution function of a performance metric called *performance profile* $\rho_o(\tau)$ to summarize the performance of different optimizers over all the tasks.

$$r_{o,a} = \frac{\max\{CPE_{o,a} : o \in \mathcal{O}\}}{CPE_{o,a}}$$
$$\rho_o(\tau) = \text{size}\{a \in \mathcal{A} : r_{o,a} \leq \tau\}/|\mathcal{A}|.$$
(2)

For tasks where a lower *CPE* is better, we just use $r_{o,a} = CPE_{o,a}/\min\{CPE_{o,a}\}$ instead to guarantee $r_{o,a} \geq 1$. The function $\rho_o(\tau)$ for all optimizers is presented in Figure 4. Based on the definition of performance profile [10], the optimizers with large probability $\rho_o(\tau)$ are to be preferred. In particular, the value of $\rho_o(1)$ is the probability that one optimizer will win over the rest and can be a reference for selecting the proper optimizer for an unknown task. We also provided a probabilistic performance profile to summarize different optimizers in Figure 2 in Appendix E.

Our findings are summarized below:

• It should be emphasized from Table 3, that under our protocol based on Hyperband, SGD performs similarly to Adam in terms of efficiency, and can even surpass it in some cases like training on

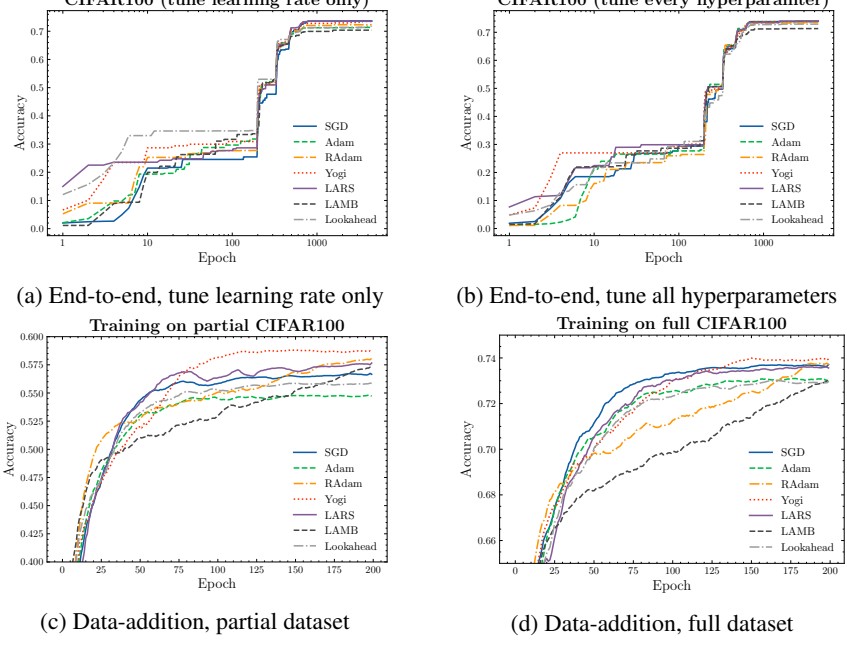

(a) End-to-end, tune learning rate only     (b) End-to-end, tune all hyperparameters

(c) Data-addition, partial dataset     (d) Data-addition, full dataset

Figure 5: End-to-end and data-addition training curves with Hyperband on CIFAR100.

CIFAR100. Under Hyperband, the best configuration of SGD is less tedious to find than random search because Hyperband can early-stop bad runs and thus they will affect less to the search efficiency and final performance.

- For image classification tasks all the methods are competitive, while adaptive methods tend to perform better in more complicated tasks (NLP, GCN, RL).
- There is no significant distinction among adaptive variants. Performance of adaptive optimizers tends to fall in the range within $1\%$ of the best result.
- According to performance profile in Figure 4, RAdam achieves probability 1 with the smallest $\tau$, and Adam is the second method achieving that. This indicates that RAdam and Adam are achieving relatively stable and consistent performance among these tasks.

### 4.2 Data-addition Training (Scenario II)

We then conduct evaluation on data-addition training based on the protocol in Algorithm 2. We choose four classification problems on CIFAR10, CIFAR100, MRPC and PPI since this data-addition training does not apply to RL. We first search the best hyperparameter configuration, denoted by $\Omega_{\text{partial}}$, under the sub training set with the ratio $\delta = 0.3$. Here we tune all hyperparameters. Then we directly apply $\Omega_{\text{partial}}$ on the full dataset for a complete training process. Training curves are shown in Figure 5c and 5d. We summarize them with *CPE* by Eq. 1 in Table 4. We have the following findings:

- There is no clear winner in data-addition training. RAdam is outperforming other optimizers in 2/4 tasks so is slightly preferred, but other optimizers except Lookahead are also competitive (within 1% range) on at least 2/4 tasks.
- To investigate whether the optimizer's ranking will change when adding 70% data, we compare the training curve on the original 30% data versus the training curve on the full 100% data in Figure 5c and 5d. We observe that the ranking of optimizers slightly changes after data addition.

## 5 Conclusions and Discussions

In conclusion, we found **there is no strong evidence that newly proposed optimizers selected in our paper consistently outperform Adam**, while each of them may be good for some particular tasks. When deciding the choice of the optimizer for a specific task, people can refer to results in Table 3. If the task is contained in Table 2, he/she can directly choose the one with the best CPE or

best peak performance based on his/her goal of the task (easiness to tune or high final performance). On the other hand, even though the desired task is not covered, people can also gain some insights from the results of the most similar task in Table 2, or refer to the performance profile in Figure 4 to pick adaptive methods like Adam. Besides choosing a suitable optimizer, our benchmarking protocol also contributes to designing a new optimizer. Using our protocol to evaluate a new optimizer can show whether it has obvious improvement over existing ones, and can serve as a routine to judge the performance of the optimizer thoroughly.

In addition to the proposed two evaluation criteria, there could be other factors that affect the practical performance of an optimizer. First, the **memory consumption** is becoming important for training large DNN models. For instance, although Lookahead performs well in certain tasks, it requires more memory than other optimizers, restricting their practical use in some memory constrained applications. Another essential criterion is the **scalability** of optimizers. When training with a massively distributed system, optimizing the performance of a large batch regime (e.g., 32K batch size for ImageNet) is of vital significance. In fact, LARS and LAMB algorithms included in our study are developed for large batch training and thus we believe scalability is an important metric worth studying in the future.

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
