# OpenReview forum: "Rethinking the Role of Hyperparameter Tuning in Optimizer Benchmarking"
_NeurIPS.cc/2021/Track/Datasets_and_Benchmarks/Round1 — Submitted to NeurIPS 2021 Datasets and Benchmarks Track (Round 1)_

### Official Review · Reviewer_64Vg · 2021-06-25
**Interesting problem with interesting observations**

**Rating:** 6
**Confidence:** 3
**Correctness:** The evaluation methods and experiment…
**Clarity:** The paper is well-written.

**Strengths:**

1.	This a very interesting problem - how much progress has recently proposed algorithms made compared with SGD or Adam.
2.	The paper is clear and easy to follow. The two new protocols are clearly described and could be easily reused by others.
3.	The experiments cover a variety of tasks, including CV, NLP, Graph, and RL. The observation is interesting and could be of interest to researchers in different domains.


**Weaknesses:**

1.	While the authors have presented results with new benchmarking protocols, they have not benchmarked the existing protocol of using the best hyperparameters. It is not clear whether the claims, such as “There is no clear winner among adaptive methods”, will still hold if all optimizers use their own best hyperparameters, or the claims can only be observed in the proposed protocols. The best hyperparameters can help us understand the performance “upper-bound” of each optimizer.
2.	Hyperband itself also has lots of hyperparameters. It is unclear how the results will be impacted by different hyperparameters of Hyperband. If the results are sensitive to the hyperparameters of Hyperband, then it would be challenging to choose a proper set of hyperparameters for a fair evaluation.


**Additional Feedback:**

A random type:

Line 150: pptimizer -> optimizer

**Documentation:**

No new dataset provided.

**Ethics:**

No concern.

**Relation To Prior Work:**

The discussion of related work is sufficient.

**Summary And Contributions:**

This paper studies benchmarking optimizers. The authors argue that the existing benchmarking protocols, i.e., using the best hyperparameters and random hyperparameter search, are either unrealistic or over-emphasizing the importance of hyperparameter tuning. Then, two new benchmarking protocols are proposed, including end-to-end training and data-addition training. The end-to-end training uses Hyperband to “smartly” search hyperparameters. The data-addition training further simulates the distribution shift. With the new evaluation protocols, the authors study how much progress has recently proposed algorithms made compared with SGD or Adam, and observe that there is no clear winner for the adaptive methods.

---

### Official Review · Reviewer_93zs · 2021-07-02
**Rethinking the Role of Hyperparameter Tuning in Optimizer Benchmarking**

**Rating:** 5
**Confidence:** 3
**Correctness:** L150; pptimizer --> optimizer

**Strengths:**

- The paper takes into consideration the practical aspects of tuning optimizers and uses a well-established gray-box hyperparameter optimization technique to independently optimize each one.

- Extensive benchmark experiments on various machine learning tasks such image classification, text classification and reinforcement learning

- An interesting investigation of the performance achieved when reusing the optimal hyperparameters on a scaled-up version of the experiment.

**Weaknesses:**

- An important aspect of the optimization process is also the training time (in terms of wall-clock time). The authors rely on the epochs as a representation of the budget, but this might be misleading because more complex optimizers require more time per epoch than the standard SGD. A small overview might be useful.

- The paper ignores some relevant optimizers, such as:
     - AdaGrad: Adaptive Subgradient Methods for Online Learning and Stochastic Optimization.
     - NovoGrad: Ginsburg, Boris, et al. "Stochastic gradient methods with layer-wise adaptive moments for training of deep networks." arXiv preprint arXiv:1905.11286 (2019).
     - SALR: Yue, Xubo, Maher Nouiehed, and Raed Al Kontar. "SALR: Sharpness-aware Learning Rates for Improved Generalization." arXiv preprint arXiv:2011.05348 (2020).


**Additional Feedback:**

In addition to reporting the duration required by each optimizer, an interesting experiment might be to investigate the impact of the  partial dataset factor \delta.

**Clarity:**

-  L142-143: Authors mention that most regions of the hyperparameter space A outperforms B. But figure 1A shows the expected loss, with optimizer B outperforming optimizer A.

- There are no details mentioned for experiments of Figure 2.

**Documentation:**

Authors use publicly available datasets.

**Relation To Prior Work:**

The authors emphasize that related work usually either over-emphasizes the role of hyperparameters, by relying on random search, or under-emphasizing by assuming optimal hyperparameters configuration. As a delineation, they propose to **hyperoptimize** the optimizers in a gray-box approach, using hyperband to eliminate configurations that perform poorly early on.

Although this is a new approach, other papers [1] have also investigated extensively the performance of optimizers under a battery of configurations in a more detail, with additional highly relevant hyperparameters such as learning rate schedules.

The conclusion, nevertheless, is consistent with all other benchmark papers.

[1] Schmidt, Robin M., Frank Schneider, and Philipp Hennig. "Descending through a Crowded Valley--Benchmarking Deep Learning Optimizers." arXiv preprint arXiv:2007.01547 (2020).

**Summary And Contributions:**

The paper benchmarks well-known optimizers on a diverse set of machine learning tasks. The authors investigate two scenarios to evaluate the performance of each optimizer: end-to-end efficiency by training the models on all training data, and data-addition efficiency, by comparing the degree of shift in performance when the optimal configuration on partially observed data is implemented on a scaled-up version.

The motivation of this paper is to investigate the performance of optimizers under the assumption that the optimal hyperparameters are unknown, a clear distinction from other benchmarks, by taking into consideration the duration of the tuning process.

The optimizers are tuned using Hyperband and evaluated using cumulative performance-early weighting scheme (CPE), and takes into account the whole trajectory of the tuning process.

The empirical evidence shows that there is no clear winner across all tasks.

---

### Official Review · Reviewer_9nW3 · 2021-07-04
**Nice Experiments, but Some Questions about Scientific Connection**

**Rating:** 5
**Confidence:** 4
**Correctness:** The specific experimental claims seem…
**Clarity:** The paper is written mostly clearly.

**Strengths:**

To the best of my knowledge, this appears to be the first benchmark that inquires into the behavior of different learning algorithms under the hyperband style hyperparameter tuning.  I think moving in this direction is potentially very useful, and so more research & benchmarks are welcome.

The total volume of experimentation seem sufficiently large, and I applaud the author's efforts in this regard.  The diversity of settings is nice to see.

**Weaknesses:**

Purpose of the Paper: I am a bit unclear on the purpose of the paper.  If the goal is to rethink the role of hyperparameter tuning, shouldn't one do a more thorough comparison with "normal" ways of doing hyperparameter tuning?  I remain unconvinced (as a function of the experiments currently in the paper) that this is the "right" way to do hyperparameter tuning.  In other words, I'm having trouble appreciating how this submission's experiments advance the science/understanding of "rethinking the role of hyperparameter tuning".  I'm very curious to hear the authors' thoughts on this issue.

Related Work Discussion: I think the discussion of context & related work could be expanded.  For instance, I am aware of one major competition/benchmark that is omitted: https://arxiv.org/abs/2104.10201  There might be others that exist as well (I didn't do a thorough literature review, but I can during the discussion period.).  At such, I think the discussion on Pages 2-3 needs to be updated.

Conclusions of Experiments: While it's fine to say that there is no clear winner, I was kind of hoping that the authors would offer more perspective and reflection on what they learned from these experiments.  I think it would have been better to spend a bit more space on this and a bit less space on setting up the problem (Pages 4-6).

Minor Comments:

-- I think Equation 1 has some formatting issues.  It took me a while to figure out what it meant.  The paragraph below Equation 1 suggests to set \lambda to a vector of all 1's, but that is in conflict with \lambda summing to 1.

-- Tables 1 & 2 & 3 & 4 are almost unreadable.

-- I would soften the language like in lines 293-294, that "newly proposed optimizers consistently outperform Adam," as that makes it seem like you compared against all the newly proposed optimizers rather than a representative sample.

**Additional Feedback:**

None.

**Documentation:**

The main piece of documentation that I'm unclear about is the type of machines used to run the experiment.

**Ethics:**

No real ethics concerns.  The datasets used are standard datasets.

**Relation To Prior Work:**

See Weaknesses section.

**Summary And Contributions:**

This paper studies benchmarking with regards to end-to-end total training/tuning time.  In other words, the amount of time it takes to train with a specific set of hyperparameters is considered and budgeted against.  The goal is to evaluate the "end-to-end" efficiency of various learning approaches.  The benchmarking is conducted on a broad range of settings, including standard vision tasks, generative modeling, NLP, and reinforcement learning.

I think the specific benchmarks conducted in this paper seem OK.  However, I'm having trouble appreciating how these experiments advances the science/understanding of "rethinking the role of hyperparameter tuning", as discussed in the Weaknesses section.

---

### Decision · Program_Chairs · 2021-07-26

**Decision:**

Reject

**Comment:**

This paper shows interesting experiments, but, as reviewer 9nW3 put it, there remains "trouble appreciating how these experiments advances the science/understanding of "rethinking the role of hyperparameter tuning".
Furthermore, the paper is not as extensive as it could be in its choice of optimizers and does not succeed in yielding actual insights beyond the fact that there is no uniformly best optimizer.
Overall, the paper is good but falls under the competitive bar of this track.